# The Pathogenic Role of Expanded CD8⁺CD28^null^ Angiogenic T Cells in ANCA-Associated Vasculitis

**DOI:** 10.3390/biomedicines13010026

**Published:** 2024-12-26

**Authors:** Haomiao Shen, Jinlin Miao, Haoyang Sun, Kui Zhang, Renli Liu, Zichao Li, Leyang Zhang, Peiyan Zhang, Jiawei Wang, Bei Zhang, Longyu Chen, Zhaohui Zheng, Ping Zhu

**Affiliations:** 1Department of Clinical Immunology of Xijing Hospital and Department of Cell Biology of National Translational Science Center for Molecular Medicine, Fourth Military Medical University, Xi’an 710032, China; shen475356312@outlook.com (H.S.); miaojinlin@fmmu.edu.cn (J.M.); sunhaoyang2223@163.com (H.S.); zhk100@fmmu.edu.cn (K.Z.); liurl24@163.com (R.L.); patricia1yn@163.com (P.Z.); vv_jevin@163.com (J.W.); zhangb122@163.com (B.Z.); longyuchen2023@163.com (L.C.); 2Department of Plastic Surgery, Xijing Hospital, Fourth Military Medical University, Xi’an 710032, China; 123lzcxjzx@fmmu.edu.cn (Z.L.); drleyang@163.com (L.Z.)

**Keywords:** angiogenic T cells, CD28^null^, pathogenesis, endothelial dysfunction

## Abstract

**Objectives:** Angiogenic T cells (Tang) are crucial in promoting angiogenesis, with the loss of CD28 serving as a marker for highly differentiated and senescent T cells. This study aims to investigate the characteristics and potential roles of CD8^+^CD28^null^ Tang in patients with ANCA-associated vasculitis (AAV). **Methods:** A cohort of AAV patients and matched healthy controls (HCs) were analyzed. Flow cytometry was used to assess the profiles of circulating CD8^+^CD28^null^ Tang. In vitro functional assays were performed to evaluate the pathogenic properties of CD8^+^CD28^null^ Tang. **Results**: CD8^+^CD28^null^ Tang levels were significantly higher in the peripheral blood of AAV patients compared to HCs, and their levels were further increased in AAV patients with MPO⁺, p-ANCA⁺, or interstitial lung disease compared to their respective control groups. Additionally, there was a positive correlation between both the percentage and absolute count of CD8^+^CD28^null^ Tang and the Birmingham Vasculitis Activity Score (BVAS). In patients with a good treatment response, both the percentage and absolute count of CD8^+^CD28^null^ Tang were significantly reduced, and this reduction was positively correlated with the decrease in BVAS scores. In vitro studies revealed that CD8^+^CD28^null^ Tang displayed enhanced chemotactic properties, induced apoptosis in human umbilical vein endothelial cells (HUVECs), and inhibited their proliferation, migration, and tube formation. **Conclusions:** AAV patients exhibit increased levels of circulating CD8^+^CD28^null^ Tang, which can be reduced following effective treatment. Furthermore, CD8^+^CD28^null^ Tang may contribute to the pathogenesis of AAV by promoting apoptosis and inhibiting the proliferation, migration, and tube formation of HUVECs.

## 1. Introduction

ANCA-associated vasculitis (AAV) is a systemic inflammatory disease of small- and medium-sized vessels, driven by autoantibodies targeting neutrophil proteins such as proteinase 3 (PR3) and myeloperoxidase (MPO) [1]. Although AAV can involve any tissue, the respiratory tract and kidneys are most frequently and severely affected [2]. Immunosuppressive therapy decreases AAV mortality; however, long-term maintenance therapy is necessary, and relapses are common [3]. Persistent coagulation and immune activation during AAV remission suggest ongoing low-grade inflammation, contributing to vascular damage and disease recurrence. Elevated circulating endothelial cells during remission further indicate sustained vascular injury [4].

Hur et al. identified angiogenic T cells (Tang) based on the expression of CD3, CD31, and CXCR4 [5]. Both animal and human studies highlight the roles of CD31^+^ T lymphocytes in capillary formation [6,7,8]. Tang cells, a specialized subset of CD3^+^ T cells, can be further classified as either CD4^+^ (CD4^+^CD31^+^CXCR4^+^) or CD8^+^ (CD8^+^CD31^+^CXCR4^+^) Tang [9,10,11]. Fluctuations in Tang cell populations have been associated with various diseases, including rheumatoid arthritis (RA), AAV, systemic sclerosis (SSc), and systemic lupus erythematosus (SLE) [12,13,14,15]. Recently, a notable increase in CD8^+^ Tang has been observed in SLE and lupus nephritis (LN) patients, particularly in those with anti-dsDNA autoantibodies [16,17]. Further investigations by Lopez et al. [12] revealed that Tang can be divided into two subsets based on CD28 expression. CD28^null^ Tang exhibits a senescent phenotype, characterized by the absence of CD28, increased CD57 levels, and a decreased expression of CD27 and CCR7. Additionally, senescent CD28^null^ Tang cells express CD56, perforin, granzyme B, and IFN-γ, suggesting these cells may have cytotoxic functions [12,18]. However, the function and specific role of CD8^+^CD28^null^ Tang in AAV patients remain inadequately understood.

Therefore, this study aims to analyze changes in CD8^+^CD28^null^ Tang in AAV patients, investigate their correlation with clinical features, and explore their potential pathogenic roles in AAV progression.

## 2. Materials and Methods

### 2.1. Participants

AAV patients were recruited from the Department of Clinical Immunology at Xijing Hospital between August 2023 and June 2024, adhering to the 2022 ACR/EULAR classification criteria [19]. A total of 63 AAV patients and 27 age- and gender-matched HCs were enrolled. The AAV patients were further categorized into MPO-AAV (*n* = 46) or PR3-AAV (*n* = 17) groups. Patients with concurrent connective tissue diseases, diabetes, pregnancy, dyslipidemia, liver disease, and malignancies were excluded from participating. Additionally, HCs were confirmed to have no pathological conditions or ongoing treatments. The demographics, clinical profiles, and current medications of the participants are summarized in Table 1. The study was approved by the Ethics Committee of Xijing Hospital and conducted in accordance with the Declaration of Helsinki. Informed consent was obtained from all participants prior to enrollment. Each participant received comprehensive information regarding the study, including its objectives, procedures, potential risks, and anticipated benefits. This process ensured that participants made an informed and voluntary decision regarding their involvement in the study. To ensure data privacy, patient information was de-identified, and each participant was assigned a unique identification code throughout data collection, analysis, and reporting. Sensitive data were encrypted and stored securely, accessible only to authorized personnel following institutional and national regulations. All analyses utilized de-identified datasets, and the research team underwent comprehensive training in data confidentiality and ethical handling procedures.

### 2.2. Flow Cytometry

To stain and analyze Tang cells and their subsets, fresh peripheral whole blood samples were labeled with the antibodies, including anti-CD3-Allophycocyanin (APC)/Cyanine7, anti-CD4-Spark UV™ 387, anti-CD8a-Fluorescein isothiocyanate (FITC), anti-CD28-Peridinin chlorophyll protein (PerCP)/Cyanine5.5, anti-CD31-Phycoerythrin (PE)/Cyanine7, and anti-CD184 (CXCR4)-Allophycocyanin (APC) (all from BioLegend, San Diego, CA, USA). Isotope-matched IgG conjugates served as control antibodies. Tang cells were identified as CD3^+^CD31^+^CD184^+^ T cells, and CD8^+^CD28^null^ Tang cells were defined as CD8^+^CD28^null^CD31^+^CD184^+^ T cells. Peripheral blood endothelial progenitor cells (EPCs, CD34^+^CD133^+^VEGFR2^+^) were stained with PE/Cyanine7-CD34, PE-CD133, and APC-CD309 (VEGFR2) antibodies (BioLegend, San Diego, CA, USA). The labeled cells were evaluated using an FACS Aria II flow cytometer (BD Biosciences, San Jose, CA, USA) and analyzed by FlowJo 7.6.1 (Tree Star, Ashland, OR, USA). Absolute cell counts were determined using Flow-Count Beads (BioLegend, San Diego, CA, USA).

### 2.3. Flow Sorting and Cell Activation

Peripheral blood mononuclear cells (PBMCs) from AAV patients were isolated via Ficoll–Paque density gradient centrifugation (Axis-Shield, Oslo, Norway) and subsequently resuspended in RPMI 1640 medium (Gibco, Grand Island, NY, USA) supplemented with 10% FBS (Gibco, Grand Island, NY, USA). The cells were stained with the following antibodies: anti-CD8a-FITC, anti-CD28-PerCP/Cyanine5.5, anti-CD31-PE/Cyanine7, and anti-CD184 (CXCR4)-APC. After staining, CD8^+^CD28^null^ Tang (CD8^+^CD31^+^CD184^+^CD28^−^) and non-Tang CD8^+^ T cells (CD8^+^CD31^−^CD184^−^ T cells) were sorted using a BD Aria III flow cytometer (BD Biosciences, San Jose, CA, USA). The purified T cells were seeded at a density of 1 × 10^5^ cells per 100 μL in 96-well plates coated with 10 μg/mL plate-bound anti-human CD3 mAb (BioLegend, San Diego, CA, USA) for 1 day to induce activation. Subsequent experiments were then performed.

### 2.4. Cell Culture

Flow cytometrically sorted Tang cells were added to HUVEC cultures under two conditions: direct co-culture and co-culture using 0.4 μm Transwell chambers (Corning Inc., Corning, NY, USA). Fresh DMEM medium (Gibco, Grand Island, NY, USA) was used to support cell growth and maintenance. Both untreated cells (negative control) and co-cultured cells were incubated for 24 h. After incubation, cells were collected and washed twice with PBS to remove suspended T cells. Apoptosis analysis of HUVECs was performed as previously described, using Annexin V-APC and PI (BioLegend, San Diego, CA, USA) staining. Apoptosis levels were subsequently analyzed using an FACS Calibur flow cytometer (BD Biosciences, San Jose, CA, USA) and FlowJo 7.6.1 software (Tree Star, Ashland, OR, USA). The same procedure was applied to human dermal microvascular endothelial cells (HDMECs) (Procell, Wuhan, China).

### 2.5. Western Blot Analysis

Cultured HUVECs were co-incubated with sorted Tang at 37 °C for 24 h. Protein concentrations were determined using a BCA kit. Total proteins were separated on a 10% SDS gel and transferred to PVDF membranes (Amersham International, Amersham, UK). Membranes were blocked in PBS containing 5% non-fat milk. Primary antibodies were incubated overnight at 4 °C, including anti-Bax (1:2000) and anti-GAPDH (1:5000) (all from Proteintech, Rosemont, IL, USA). Membranes were washed with TBST and then incubated with HRP-conjugated secondary antibodies (Beyotime, Shanghai, China). Bands were visualized with the ECL system (Perkin Elmer, Waltham, MA, USA). Semi-quantitative analysis of scanned films was performed using Quantity One-4.6.2 (Bio-Rad, Segrate, Italy), using GAPDH as a loading control to determine the relative expression levels of target proteins. All Western blot experiments were independently repeated three times.

### 2.6. Chemotaxis Analysis

The chemotaxis of CD8^+^CD28^null^ Tang towards HUVECs was evaluated using a 3 μm pore size Transwell system (Corning Inc., Corning, NY, USA). The top chamber of the Transwell system was loaded with CD8^+^CD28^null^ Tang or non-Tang CD8^+^ T cells, while HUVECs were cultured in the lower wells of the Transwell system. After being incubated for 24 h, the cells were extracted from the upper and lower wells. Then, to distinguish between the two cell types, anti-CD8, anti-CD28, anti-CD31, and anti-CD184 antibodies were used for staining, followed by flow cytometry for counting CD8^+^CD28^null^ Tang and non-Tang CD8^+^ T cells. The percentage of each cell type migrating to the lower chamber relative to the total number of cells introduced was calculated.

### 2.7. 2D Migration–Scratch Assay

A scratch assay was performed to assess cell migration and motility. In brief, 1 × 10^4^ HUVECs were seeded into 24-well plates and cultured to nearly 100% confluence. To create artificial wounds for the migration studies, a 200 µL pipette tip was used to scratch the cell monolayers. The monolayers were then washed twice with PBS to remove dislodged cells. Subsequently, 1 mL of serum-free ECM medium was added to each well (ScienCell, Carlsbad, CA, USA), along with 1 × 10^5^ CD8^+^CD28^null^ Tang or non-Tang CD8^+^ T cells. Images of the scratches were captured at time points of 0 h and 12 h post-incubation at 37 °C. To ensure comprehensive coverage of the wound area in each well, three images per scratch were taken. Cell migration into the wound was quantified using the area measurement tool in ImageJ 1.52a (NIH, Bethesda, MD, USA), which measured the cell-free regions to determine migration efficiency.

### 2.8. Cell Counting Kit-8 (CCK-8) Assay

After 24 h of co-culture, HUVECs were seeded at a density of 2 × 10^3^ cells/mL in 96-well plates. Each well received 100 µL of cell culture medium and CCK-8 reagent (Beyotime, Shanghai, China), followed by incubation for 2 h at 37 °C. Absorbance was measured at 450 nm using a microplate reader (Thermo Fisher Scientific, Waltham, MA, USA). Triplicate wells were set for each treatment group, and the average absorbance was calculated. Cell viability curves were plotted with time on the *x*-axis and optical density (OD) on the *y*-axis. Cell viability (%) was calculated using the formula: Cell viability (%) = (Absorbance of experimental group/Absorbance of control group) × 100%. Experiments were independently repeated three times.

### 2.9. 5-Ethynyl-2′-deoxyuridine (Edu) Assay

The Edu assay was performed using the BeyoClick™ Edu Cell Proliferation Kit with Alexa Fluor 594 (Beyotime, Shanghai, China). The cells were initially washed with PBS and then incubated with Edu solution for 2 h. Subsequently, nuclei were stained with DAPI solution (Beyotime, Shanghai, China). After washing, samples were observed under an Olympus inverted microscope (Nikon, Tokyo, Japan) for quantification of cell proliferation.

### 2.10. Tube Formation Assay

To assess the tube formation capability of HUVECs, 200 μL of Matrigel (BD Biosciences, San Jose, CA, USA) was added to each well of a 24-well plate (Corning Inc., Corning, NY, USA) and allowed to solidify at room temperature for one hour. Subsequently, HUVECs were suspended at a density of 1 × 10^5^ cells per well, combined with the appropriate number of Tang cells, and added to each well. The cells were then incubated at 37 °C for six hours. Tube formation was evaluated and quantified using an ECLIPSE-Ti inverted microscope (Nikon, Tokyo, Japan), and the total tube length was measured to determine the tube formation capacity of HUVECs.

### 2.11. Statistical Analysis

Data are presented as mean ± standard deviation and analyzed using GraphPad Prism 9 (GraphPad Software, La Jolla, CA, USA). Statistical analyses were performed with SPSS 20.0 (IBM, Armonk, NY, USA). Differences between groups were assessed using the non-parametric Mann–Whitney U test and paired *t*-test. Spearman rank correlation was used to evaluate correlations. A *p*-value of less than 0.05 was considered statistically significant.

## 3. Results

### 3.1. Increased Percentage of Circulating CD8^+^CD28^null^ Tang in AAV Patients

Flow cytometry was utilized to analyze the levels of Tang subsets in patients with AAV and HCs (Figure 1A). As shown in Figure 1B, the percentage of circulating CD3^+^CD28^null^ Tang was significantly higher in AAV patients compared to HCs; however, there was no significant difference in cell counts between the two groups (Figure 1E). Additionally, the percentage and cell count of CD4^+^CD28^null^ Tang in AAV patients were not significantly different from those in HCs (Figure 1C,F). In contrast, both the percentage and cell count of CD8^+^CD28^null^ Tang were significantly elevated in AAV patients compared to HCs (Figure 1D,G).

### 3.2. Correlation Analysis of CD8^+^CD28^null^ Tang and Clinical Features of AAV Patients

Subsequently, CD8^+^CD28^null^ Tang cells were selected for a correlation analysis using the clinical features of the AAV patients. We observed that patients with MPO^+^, p-ANCA^+^, ILD^+^, and BVAS > 15 exhibited significantly higher percentages and cell counts of CD8^+^CD28^null^ Tang compared to those with MPO^−^, p-ANCA^−^, ILD^−^, and BVAS ≤ 15, respectively (Figure 2A–F). In contrast, there was no statistical difference in the percentages or cell counts of CD8^+^CD28^null^ Tang between patients with ESR^+^ and ESR^−^, or between those with CRP^+^ and CRP^−^ (Figure 2E,F). A further analysis demonstrated that the percentage and cell count of CD8^+^CD28^null^ Tang were positively correlated with AAV disease activity (BVAS), but not with ESR or CRP (Figure 2G–L).

### 3.3. Comparison of CD8^+^CD28^null^ Tang in AAV Patients with Different Treatment Responses

The levels of CD8^+^CD28^null^ Tang were further analyzed in relation to the treatment response of AAV patients. AAV patients with a BVAS score > 15 were classified as being in the active phase, while those with BVAS ≤ 15 were categorized as being in the stable phase [20,21,22]. Based on this classification, eight patients who exhibited a good treatment response (transitioning from the active to stable phase within one month) and eight patients who showed a moderate response (showing persistent activity over a one-month interval) were selected [22,23,24]. The clinical characteristics are summarized in Table 2, with no significant differences observed between the two groups. The results indicated that the levels of ESR, CRP, BVAS, and CD8^+^CD28^null^ Tang were decreased in AAV patients classified as having had a good treatment response (Figure 3A). In contrast, these parameters showed no significant changes in patients with persistent activity (a moderate treatment response) (Figure 3B). A further analysis revealed a positive correlation between reductions in ESR, CRP, and BVAS and the decrease in CD8^+^CD28^null^ Tang (Figure 3C,D). These findings suggest that CD8^+^CD28^null^ Tang may play a pathogenic role in AAV and may serve as a biomarker for patient treatment response.

### 3.4. Correlation Between Circulating EPC and CD8^+^CD28^null^ Tang Levels in AAV Patients

The levels of endothelial progenitor cells (EPCs; CD34^+^CD133^+^VEGFR^+^) in AAV patients were assessed by flow cytometry (Figure 4A). Compared to HCs, AAV patients exhibited a significant reduction in both the percentage and cell count of EPCs (Figure 4B), consistent with previous studies [14]. Furthermore, the percentage and cell count of CD8^+^CD28^null^ Tang were negatively correlated with both the percentage and cell count of EPCs, respectively (Figure 4C). These findings suggest that CD8^+^CD28^null^ Tang may significantly impact vascular endothelial cells, potentially contributing to the development of AAV. Furthermore, a Transwell assay was also conducted to investigate the differential chemotactic effects of CD8^+^CD28^null^ Tang and non-Tang CD8^+^ T cells on vascular endothelial cells (Figure 4D). After 24 h, a higher transmigration rate of CD8^+^CD28^null^ Tang compared to non-Tang CD8^+^ T cells was observed (Figure 4E). This suggests that CD8^+^CD28^null^ Tang exhibit enhanced chemotactic capabilities toward HUVECs compared to non-Tang CD8^+^ T cells.

### 3.5. CD8^+^CD28^null^ Tang Cells Induce HUVEC Apoptosis in Vitro

The effects of CD8^+^CD28^null^ Tang on HUVEC apoptosis were investigated in vitro. After a 24 h co-culture, the HUVECs exposed to CD8^+^CD28^null^ Tang exhibited a significantly higher rate of apoptosis compared to the control group and HUVECs co-cultured with non-Tang CD8^+^ T cells (Figure 5A). A Western blot analysis further confirmed a significant increase in the levels of the pro-apoptotic protein Bax in HUVECs co-cultured with CD8^+^CD28^null^ Tang, whereas Bax levels remained unchanged in the non-Tang CD8^+^ T cell and control groups (Figure 5B). To explore the mechanism of CD8^+^CD28^null^ Tang-induced HUVEC apoptosis, a Transwell co-culture system was used. The results showed that separating the cells with a Transwell membrane significantly reduced HUVEC apoptosis (Figure 5C). These findings indicate that CD8^+^CD28^null^ Tang exert a contact-dependent cytotoxic effect on endothelial cells.

To determine whether these effects were specific to HUVECs, HDMECs were also subjected to the same experimental conditions. Similar to HUVECs, HDMECs showed a significantly higher rate of apoptosis when co-cultured with CD8^+^CD28^null^ Tang compared to co-cultures with non-Tang CD8^+^ T cells or under control conditions (Appendix A). These results highlight the broad cytotoxic potential of CD8^+^CD28^null^ Tang toward various types of endothelial cells.

### 3.6. CD8^+^CD28^null^ Tang Cells Inhibit HUVEC Proliferation, Migration, and Tube Formation

To assess the effects of CD8^+^CD28^null^ Tang on the biological functions of HUVECs, we examined their effects on HUVEC proliferation, migration, and tube formation. First, using the EdU incorporation assay, we observed a significant reduction in the proportion of EdU-positive HUVECs in the group co-cultured with CD8^+^CD28^null^ Tang, compared to the control and non-Tang CD8⁺ T cell co-culture groups (Figure 6A,B). This suggests a marked inhibition of HUVEC proliferation. The CCK-8 assay further confirmed these findings, showing a significant suppression of HUVEC proliferation in the CD8^+^CD28^null^ Tang group, with statistically significant differences relative to the control and non-Tang CD8^+^ T cell groups (Figure 6C). Next, in the scratch assay, CD8^+^CD28^null^ Tang significantly inhibited HUVEC migration. The wound healing area was notably reduced in the group co-cultured with CD8^+^CD28^null^ Tang compared to both the control group and the non-Tang CD8^+^ T cell co-culture group (Figure 6D), reinforcing the inhibitory effect of CD8^+^CD28^null^ Tang on endothelial cell migration. Finally, the tube formation assay revealed a significant decrease in the total tube length formed by HUVECs in the presence of CD8^+^CD28^null^ Tang, compared to the control and non-Tang CD8^+^ T cell groups, indicating a severe impairment in tube formation capacity (Figure 6E). These findings suggest that CD8^+^CD28^null^ Tang may contribute to the pathogenesis of AAV by inhibiting endothelial cell proliferation, migration, and tube formation.

## 4. Discussion

In AAV, vascular abnormalities occur early, and angiogenesis is compromised from the disease onset [25]. Tang cells may regulate endothelial cell proliferation and function, thereby contributing to vascular repair in AAV [14]. Similar observations have been noted in SSc [26]. Tang cells can also express either CD4 or CD8 and can be categorized into two subsets based on CD28 expression [12]. In this study, we focus on the recently identified CD8^+^CD28^null^ Tang in AAV patients. Our data show that both the percentage and cell count of circulating CD8^+^CD28^null^ Tang are elevated in AAV patients compared to HCs. CD8^+^CD28^null^ Tang levels are further elevated in AAV patients with MPO^+^, p-ANCA^+^, interstitial lung disease, and active disease status. Notably, CD8^+^CD28^null^ Tang levels significantly decreased following a good treatment response, and the degree of reduction was positively correlated with improvements in disease activity. Furthermore, our in vitro work demonstrates that CD8^+^CD28^null^ Tang cells inhibit HUVEC proliferation, migration, and tube formation and contribute to the increased apoptosis of HUVECs and HDMECs. In summary, our results suggest that CD8^+^CD28^null^ Tang cells play a potential pathogenic role in AAV.

Tang cells have been extensively studied in various autoimmune rheumatic diseases, although their expression patterns vary considerably. Increased levels of circulating Tang have been reported in LN, SSc, and primary Sjögren’s syndrome (pSS) patients [9,11,16]. Conversely, reduced Tang levels have been reported in Behçet’s disease (BD), particularly during heightened disease activity, indicating compromised vascular repair capabilities [27]. Interestingly, the performance of Tang levels in RA is inconsistent, including some reports of increased Tang levels, but other studies show decreased Tang levels in RA, which suggests that Tang cells play a complex role in angiogenesis and the pathogenesis of RA [13,28]. Furthermore, while Tang cells are traditionally classified by CD3, CD31, and CXCR4, their functional diversity is more clearly defined when further categorized by CD4, CD8, and CD28 expression. These profiles and functional heterogeneity suggest that specific Tang subsets may contribute differentially to the pathogenesis of various autoimmune conditions.

Our study, along with existing research, emphasizes the diverse roles of Tang subsets in autoimmune diseases, particularly in AAV. In SLE, CD8⁺ Tang levels are significantly elevated, especially in patients with anti-dsDNA antibodies, indicating their involvement in endothelial damage and an increased cardiovascular risk [17]. Notably, CD28^null^ Tang cells, which exhibit cytotoxic and senescent features, are associated with inflammatory markers such as TNF-α and IFN-α, suggesting a possible pathogenic role [12]. In our study on AAV, we also observed elevated levels of CD8⁺CD28^null^ Tang, particularly in patients experiencing severe vascular complications, which supports their involvement in vascular injury via inflammatory mechanisms, similar to the findings in SLE [12]. Furthermore, the levels of CD8⁺CD28^null^ Tang in AAV patients correlate with BVAS, indicating their potential as biomarkers for disease severity.

As previously reported, CD4⁻CD8⁻CD28^null^ Tang levels correlate with increased disease activity and EPC levels in pSS, implicating these cells in endothelial dysfunction and impaired vascular repair [11]. Similarly, we found that elevated CD8⁺CD28^null^ Tang levels are correlated with increased disease activity and decreased EPC counts in AAV, reinforcing their role in endothelial dysregulation and disease progression. Additionally, we observed that CD8⁺CD28^null^ Tang levels are elevated in AAV patients with high BVAS and ILD, suggesting their involvement in promoting vascular injury rather than facilitating repair. However, no correlation was found between CD8⁺CD28^null^ Tang levels and traditional inflammatory markers, such as ESR or CRP. Recent studies suggest that these markers may not accurately reflect disease activity or severity in AAV [29,30], as the progression of the disease involves complex, localized immune responses that these markers may not fully capture. Therefore, relying solely on a single inflammatory marker may be insufficient for accurately assessing disease severity. Additionally, our study demonstrated that successful treatment led to a significant reduction in CD8⁺CD28^null^ Tang levels, which positively correlated with a decreased BVAS. Overall, these findings indicate that specific Tang subsets, particularly CD8⁺CD28^null^ Tang, may serve as promising biomarkers for disease activity and therapeutic response, contributing to the development of endothelial dysfunction. These findings also highlight their potential as causative agents and therapeutic targets in AAV. To further validate these results, larger-scale studies with a broader patient population, including untreated individuals or those receiving a placebo, and long-term follow-up observations are necessary. The primary limitation of this study is the limited sample size and the potential introduction of confounding bias due to differences in treatment types. Although efforts have been made to control these biases through patient selection and a subgroup analysis, a comprehensive stratified analysis of the specific effects of different treatment types has not yet been performed. Therefore, future research should incorporate larger sample sizes and adopt stratified analysis methods to more accurately elucidate the potential impact of treatment on immune biomarkers.

To distinguish the specific effects of CD8⁺CD28^null^ Tang from those of non-Tang CD8⁺ T cells on endothelial damage, we performed a Transwell chemotaxis assay. The results revealed that CD8^+^CD28^null^ Tang exhibited enhanced migratory capabilities toward vascular endothelial cells compared to non-Tang CD8^+^ T cells. This finding suggests that CD8^+^CD28^null^ Tang may play a more prominent role in inflammatory responses and interactions with endothelial cells, thereby contributing to endothelial dysfunction and vascular damage. Based on these observations, we hypothesize that CD8⁺CD28^null^ Tang cells contribute to AAV pathogenesis by targeting and damaging vascular endothelial cells. To test this hypothesis, we co-cultured CD8⁺CD28^null^ Tang with HUVECs in vitro. The experiments demonstrated that CD8⁺CD28^null^ Tang cells induce apoptosis in HUVECs in vitro through a contact-dependent mechanism, while simultaneously inhibiting their proliferation, migration, and tube formation. Previous studies suggest that the receptor profile of CD28^null^ T cells is altered, potentially decreasing their reliance on classical MHC recognition [31]. For instance, the activation of these cells may involve non-MHC activating receptors, such as CD16, CD161, or CD244, which are frequently upregulated on CD28^null^ T cells [32,33,34]. In addition, several adhesion molecules, including CD2, are upregulated on CD28⁻ T cells, contributing to their stable adhesion which likely enhances the precision and efficiency of cytotoxic molecule delivery, such as perforin and granzyme, thereby increasing cytotoxic efficacy [18,33]. Therefore, we propose that CD8⁺CD28^null^ Tang cells may not exclusively rely on TCR-MHC I interactions to mediate their cytotoxic effects on HUVECs. The further elucidation of the molecular pathways involved in CD8⁺CD28^null^ Tang-mediated endothelial chemotaxis and damage is essential for the development of targeted therapies to mitigate vascular complications in autoimmune diseases.

## 5. Conclusions

In conclusion, our study demonstrates a significant expansion of circulating CD8^+^CD28^null^ Tang cells in AAV patients, which decrease following a good treatment response and are positively correlated with reduced disease activity. Moreover, CD8^+^CD28^null^ Tang cells exhibit substantial endothelial-damaging effects in vitro by promoting apoptosis and inhibiting the proliferation, migration, and tube formation of HUVECs. These findings suggest that CD8^+^CD28^null^ Tang cells may serve as valuable biomarkers for disease severity and treatment response, underscoring their pathogenic role in vascular injury in AAV. Consequently, targeting CD8^+^CD28^null^ Tang may offer potential therapeutic strategies to mitigate endothelial dysfunction in autoimmune conditions.

## Figures and Tables

**Figure 1 biomedicines-13-00026-f001:**
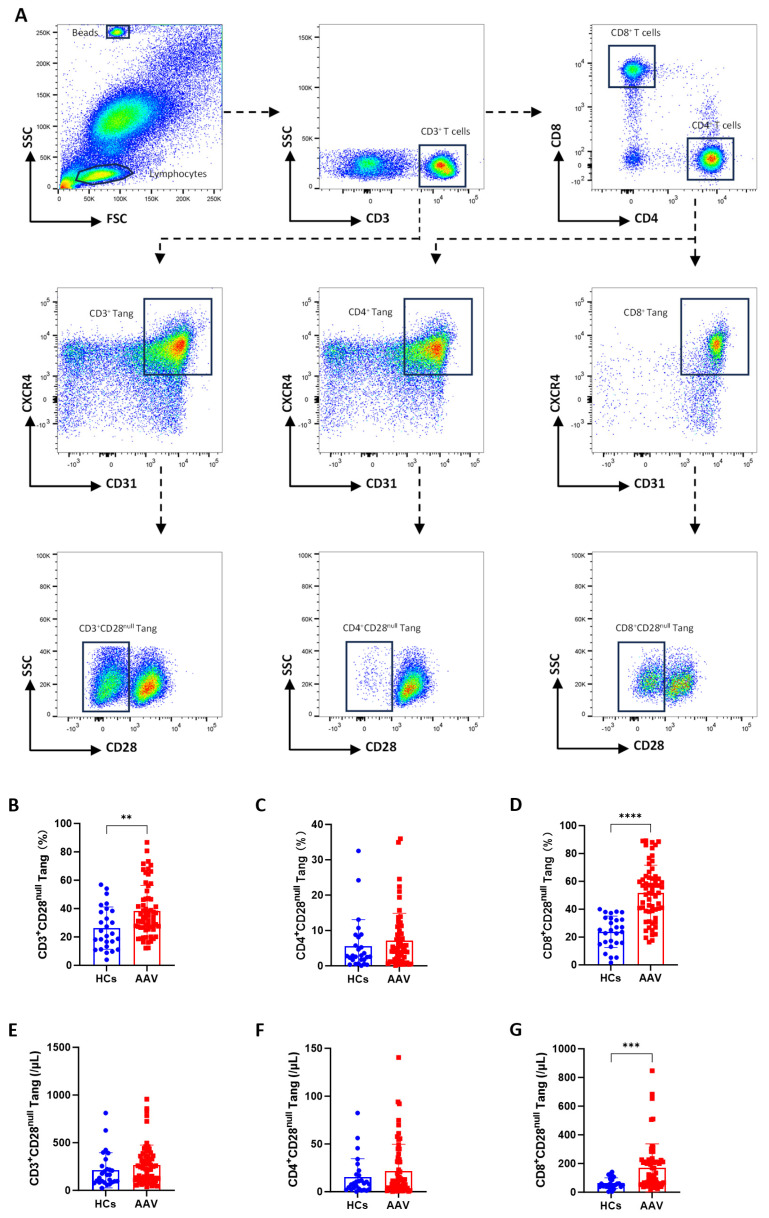
Phenotypic analysis of circulating CD8^+^CD28^null^ Tang in AAV patients. (**A**) Gating strategy for identifying CD3^+^, CD4^+^, and CD8^+^CD28^null^ Tang. The boxes indicate the gated populations and the colors represent cell density, with warmer colors (e.g., red and yellow) indicating higher density, and cooler colors (e.g., blue) indicating lower density. (**B**–**D**) Percentages of CD3^+^CD28^null^ Tang, CD4^+^CD28^null^ Tang, and CD8^+^CD28^null^ Tang in AAV patients (*n* = 63) and healthy controls (HCs, *n* = 27). (**E**–**G**) Absolute counts of CD3^+^CD28^null^ Tang, CD4^+^CD28^null^ Tang, and CD8^+^CD28^null^ Tang per microliter of whole blood in AAV patients (*n* = 63) and HCs (*n* = 27). ** *p* < 0.01; *** *p* < 0.001; **** *p* < 0.0001.

**Figure 2 biomedicines-13-00026-f002:**
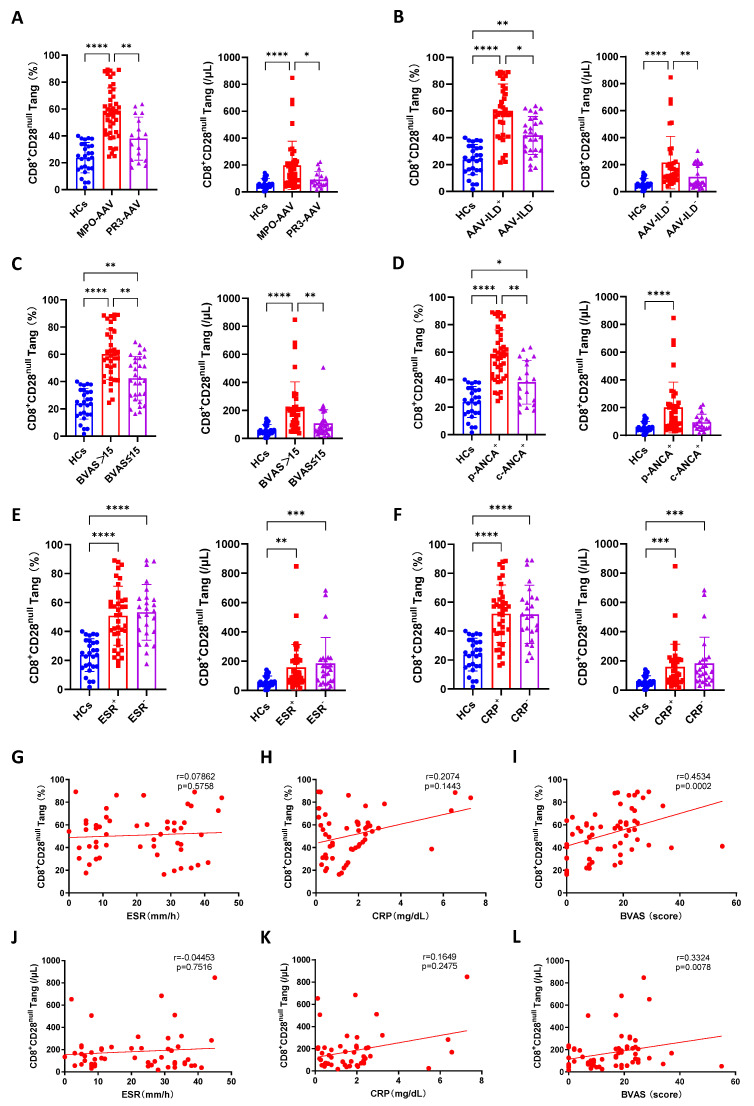
Correlation of circulating CD8^+^CD28^null^ Tang and clinical features in AAV patients. (**A**–**F**) Percentage and cell count of CD8^+^CD28^null^ Tang in HCs and in AVV patients with MPO+ (**A**), ILD+ (**B**), BVAS > 15 (**C**), p-ANCA+ (**D**), elevated ESR (**E**), elevated CRP (**F**). (**G**–**L**) Correlation between ESR, CRP, and BVAS with the percentage and absolute cell count of CD8^+^CD28^null^ Tang in AAV patients (*n* = 63). * *p* < 0.05; ** *p* < 0.01; *** *p* < 0.001; **** *p* < 0.0001.

**Figure 3 biomedicines-13-00026-f003:**
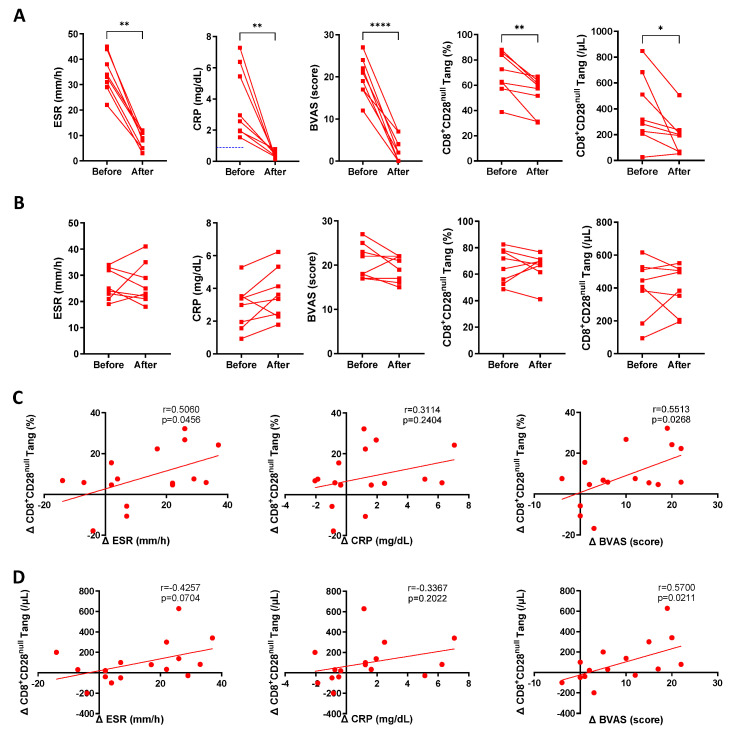
Levels of CD8^+^CD28^null^ Tang in AAV patients before and after follow-up. (**A**) Changes in ESR, CRP, BVAS, and percentage and count of CD8^+^CD28^null^ Tang before and after follow-up in AAV patients (*n* = 8) with good treatment response. (**B**) Changes in ESR, CRP, BVAS, and percentage and count of CD8^+^CD28^null^ Tang before and after follow-up in AAV patients (*n* = 8) with moderate treatment response. (**C**) Correlation between the changes in CD8^+^CD28^null^ Tang percentage and changes in ESR, CRP, and BVAS in AAV patients before and after follow-up (*n* = 16). (**D**) Correlation between the changes in CD8^+^CD28^null^ Tang cell count and changes in ESR, CRP, and BVAS in AAV patients before and after follow-up (*n* = 16). * *p* < 0.05; ** *p* < 0.01; **** *p* < 0.0001.

**Figure 4 biomedicines-13-00026-f004:**
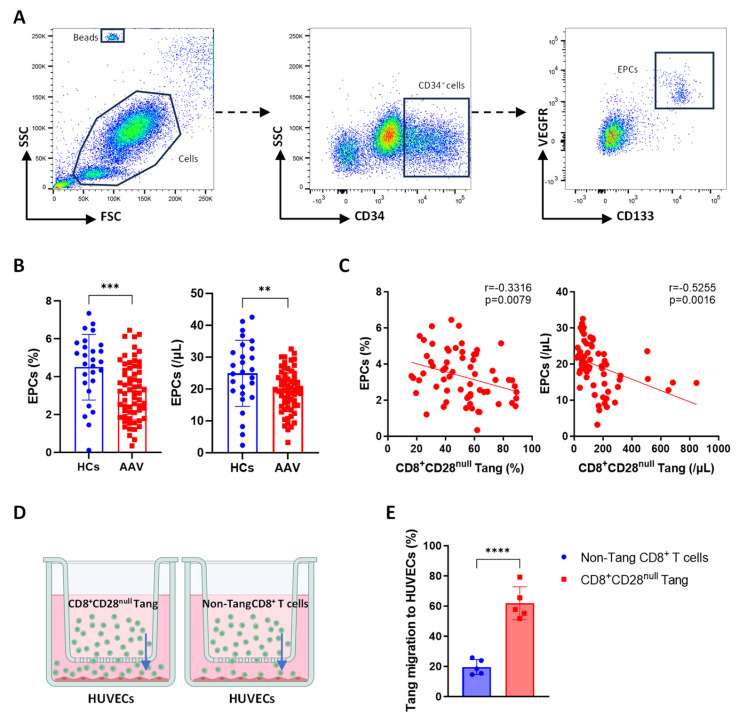
Circulating EPC levels and their correlation with CD8^+^CD28^null^ Tang in AAV patients. (**A**) The gating strategy for flow cytometric analysis of EPCs. The boxes indicate the gated populations and the colors represent cell density, with warmer colors (e.g., red and yellow) indicating higher density, and cooler colors (e.g., blue) indicating lower density. (**B**) Percentage and absolute count of EPCs in AAV patients (*n* = 63) and HCs (*n* = 27). (**C**) Analysis of the correlation between EPC percentage and CD8^+^CD28^null^ Tang percentage, as well as between the absolute counts of both in AAV patients. (**D**) Chemotaxis of CD8^+^CD28^null^ Tang and non-Tang CD8^+^ T cells from AAV patients towards HUVECs was evaluated using a 3 μm pore size Transwell system. Arrows indicate the direction of cell chemotaxis. (**E**) After 24 h, flow cytometry was used to calculate the percentage of CD8^+^CD28^null^ Tang and non-Tang CD8^+^ T cells in the lower wells. ** *p* < 0.01; *** *p* < 0.001; **** *p* < 0.0001.

**Figure 5 biomedicines-13-00026-f005:**
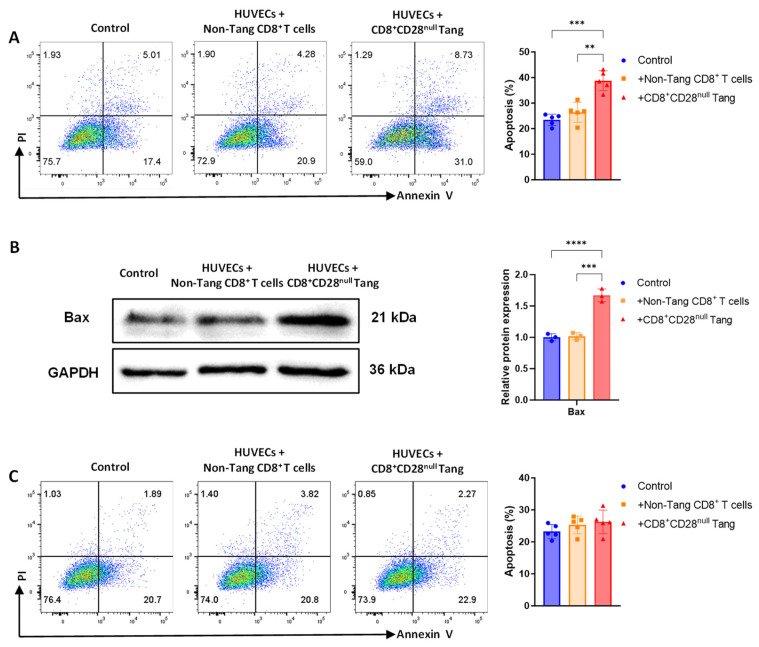
Induction of apoptosis in HUVECs by CD8^+^CD28^null^ Tang from AAV patients. (**A**) Annexin V/PI staining was used to assess HUVEC apoptosis after 24 h co-culture with CD8^+^CD28^null^ Tang and non-Tang CD8^+^ T cells from AAV patients. HUVECs alone served as the control group (Control). (**B**) Western blot analysis of apoptosis-related protein Bax expression in HUVECs after 24 h co-culture with CD8^+^CD28^null^ Tang and non-Tang CD8^+^ T cells from AAV patients. HUVECs cultured alone serve as the control group (Control). (**C**) Annexin V/PI staining was used to assess apoptosis in HUVECs after Transwell co-culture with CD8^+^CD28^null^ Tang and non-Tang CD8^+^ T cells from AAV patients. HUVECs cultured alone served as the control group (Control). The colors represent cell density, with warmer colors (e.g., red and yellow) indicating higher density, and cooler colors (e.g., blue) indicating lower density. ** *p* < 0.01; *** *p* < 0.001; **** *p* < 0.0001.

**Figure 6 biomedicines-13-00026-f006:**
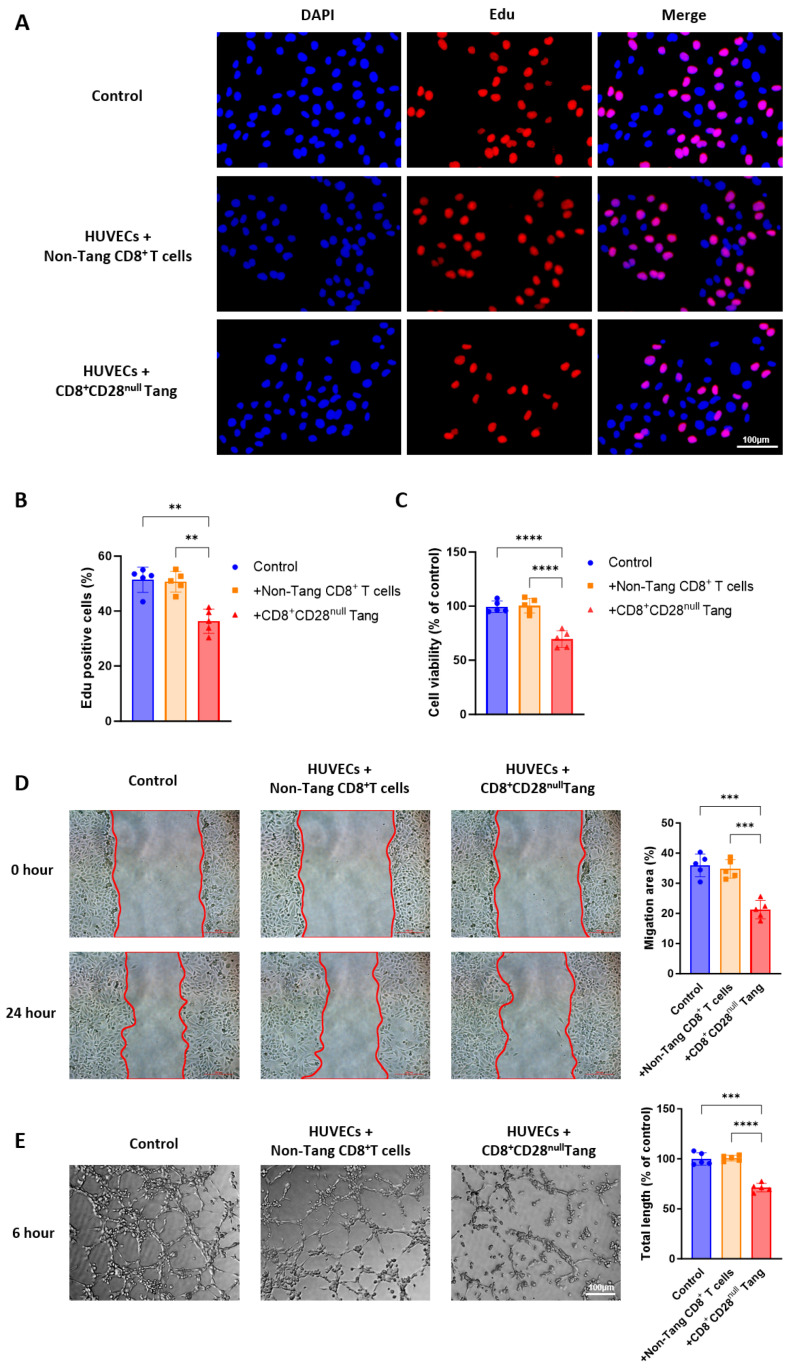
CD8^+^CD28^null^ Tang from AAV patients inhibits the proliferation and migration of HUVECs. (**A**–**C**) The proliferation of HUVECs after co-culture with CD8^+^CD28^null^ Tang and non-Tang CD8^+^ T cells from AAV patients was assessed using Edu (**A**,**B**) and CCK8 (**C**) assays. HUVECs cultured alone served as the blank control (Control). (**D**) The migration capacity of HUVECs was assessed after co-culture with CD8^+^CD28^null^ Tang and non-Tang CD8^+^ T cells from AAV patients. HUVECs alone served as the control group (Control). The red lines indicate the boundaries of cell migration. (**E**) The tube formation ability of HUVECs was assessed after co-culture with CD8^+^CD28^null^ Tang and non-Tang CD8^+^ T cells from AAV patients. HUVECs cultured alone served as the blank control (Control). The scale bar represents 100 μm (**A**,**D**,**E**). ** *p* < 0.01; *** *p* < 0.001; **** *p* < 0.0001.

**Table 1 biomedicines-13-00026-t001:** Characteristics of the AAV patients and HCs.

Characteristics	AAV (*n* = 63)	HCs (*n* = 27)
Demographic information		
Age (years), mean (SD)	55.3 ± 15.23	52.5 ± 14.32
Female (%)	33 (52.4%)	14 (51.9%)
Vasculitis type		
MPA, *n* (%)	55 (87.3%)	NA
GPA, *n* (%)	8 (13.7%)	NA
Duration of disease, m (median, IQR)	63 (31, 153)	NA
Laboratory examinations		
Proteinuria, g/24 h (median, IQR)	1.01 (0.61, 1.83)	NA
MPO (positive)	46 (73.0%)	NA
PR3 (positive)	17 (27.0%)	NA
p-ANCA (positive)	44 (69.8%)	NA
c-ANCA (positive)	19 (30.2%)	NA
ESR, mm/h (median, IQR)	20 (8, 33)	NA
CRP, mg/dL (median, IQR)	1.43 (0.45, 2.31)	NA
Serum C3, mg/dL (mean, SD)	85.32 ± 28.45	NA
Serum C4, mg/dL (mean, SD)	22.83 ± 11.34	NA
IgA, mg/dL (mean, SD)	255.68 ± 114.89	NA
IgG, g/dL (mean, SD)	1422.42 ± 488.76	NA
IgM, mg/dL (mean, SD)	111.23 ± 68.32	NA
BVAS (mean, SD)	14.56 ± 11.55	NA
Comorbidities		
Hypertensive diseases, *n* (%)	20 (31.7%)	NA
Lung diseases, *n* (%)	35 (55.6%)	NA
Type 2 diabetes mellitus, *n* (%)	5 (8.0%)	NA
Coronary heart disease, *n* (%)	5 (8.0%)	NA
Treatment, *n* (%)		
None or NSAIDs	7 (11.1%)	NA
Glucocorticoids	49 (77.8%)	NA
Immunosuppressive drugs	35 (55.6%)	NA
Biologics	14 (22.2%)	NA

MPA, microscopic polyangiitis; GPA, granulomatosis with polyangiitis; MPO, myeloperoxidase; PR3, proteinase 3; p-ANCA, perinuclear anti-neutrophil cytoplasmic antibodies; c-ANCA, cytoplasmic anti-neutrophil cytoplasmic antibodies; ESR, erythrocyte sedimentation rate; CRP, C-reactive protein; BVAS, Birmingham Vasculitis Activity Score; NSAIDs, nonsteroidal anti-inflammatory drugs; NA, not applicable.

**Table 2 biomedicines-13-00026-t002:** Clinical characteristics of AAV patients based on treatment response.

Characteristics	Good Response (*n* = 8)	Moderate Response (*n* = 8)
Age (years), mean (SD)	58.63 ± 12.18	56.25 ± 7.17
Female (%)	4 (50.0%)	3 (37.5%)
Laboratory examinations		
MPO (positive, %)	7 (87.5%)	6 (75%)
PR3 (positive, %)	1 (12.5%)	2 (25%)
p-ANCA (positive, %)	7 (87.5%)	6 (62.5%)
c-ANCA (positive, %)	1 (12.5%)	2 (25%)
Medication regimen (Last 90 Days)		
Glucocorticoids (%)	8 (100%)	8 (100%)
- Prednisone acetate	6 (75%)	5 (62.5%)
- Prednisolone	2 (25%)	3 (37.5%)
Cyclophosphamide (%)	4 (50%)	5 (75%)
- MTX	1 (12.5%)	2 (25%)
- MMF	3 (37.5%)	3 (37.5%)
Biologics (%)	3 (37.5%)	2 (20%)
- Rituximab	3 (37.5%)	2 (20%)
Follow-up time (days), median (IQR)	47.0 (39.5, 56.25)	52 (42.5, 61.5)

MPO, Myeloperoxidase; PR3, proteinase 3; p-ANCA, perinuclear anti-neutrophil cytoplasmic antibodies; c-ANCA, cytoplasmic anti-neutrophil cytoplasmic antibodies; MTX, methotrexate; MMF, mycophenolate mofetil.

## Data Availability

Enquiries regarding the original contributions presented in the study can be directed to the corresponding author.

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
