# Peer review of "The Pathogenic Role of Expanded CD8⁺CD28null Angiogenic T Cells in ANCA-Associated Vasculitis"

_biomedicines, 2024, doi:10.3390/biomedicines13010026_

Round 1
Reviewer 1 Report
Comments and Suggestions for Authors
The authors run a "correlation" study, where they tried to link amount of CD8+CD28null Tang in the blood of AAV patients with the disease severity. These data were supported with the in vitro experiments, in which they used a basic system (co-culture of CD8+CD28null Tang with HUVECs) and determined apoptosis, migration and proliferation of HUVECs.
Overall, the study is performed well. As I'm not a medical doctor, I was mainly focusing on the in vitro experiments. Therefore, I would suggest a couple of things to improve the data and conclusions.
1. HUVECs are quite sensitive to various treatments in vitro. Therefore, I would recommend to reproduce the authors' data using other type(s) of ECs, such as dermal microvascular ECs (from PromoCell, for example), which would be more relevant for the authors' study. I think at least the apoptotic effects should be shown using this type of ECs.
2. I would suggest a tube-formation assay in vitro. This is more functional assay for the ECs, compared to the proliferation (EdU) or apoptosis assays.
3. Can the authors show somehow what would be the mechanism of biological activity of those CD8+CD28null Tang cells? I mean do they need cell-cell contacts or they perform their actions via secreted cytokines?
4. In Discussion, the authors should discuss whether the CD8+CD28null Tang cells do recognize major immune complexes on target cells (MHC) and therefore they get activated or inhibited? Or what does causes their activation?
5. Is there mouse model of AAV?
Minor comment:
Figure 4D: Number of migrated Tang cells should be visually noticeable.
Reviewer 2 Report
Comments and Suggestions for Authors
The study explores a relatively underexamined subset of T cells, CD8⁺CD28null angiogenic T cells, in the context of ANCA-associated vasculitis (AAV). This focus could contribute significantly to understanding the disease’s pathogenesis. However, there is a list of comments:
The study’s participant cohort lacks diversity (all patients from one hospital in China). This limitation raises concerns about the generalizability of the findings to global populations with varying genetic backgrounds and environmental exposures.
The absence of untreated or placebo groups limits the understanding of baseline conditions and potential confounding variables.
The sample size (63 AAV patients, 27 HCs) is relatively small for a study making broad claims about biomarkers and treatment efficacy.
Some statistical tests (e.g., correlations with ESR and CRP) fail to show significance. These results are not adequately discussed, which could indicate gaps in the proposed mechanisms.
The article does not elaborate on how informed consent was obtained or how data privacy was maintained. While ethics committee approval is mentioned, a clearer explanation would enhance transparency.
The treatments analyzed (e.g., glucocorticoids and immunosuppressants) are not sufficiently detailed, and their impacts on CD8⁺CD28null Tang levels are not independently assessed.
Differences between treatment types might confound results.
Comments on the Quality of English LanguageThe English could be improved to more clearly express the research.
Round 2
Reviewer 1 Report
Comments and Suggestions for Authors
The authors sufficiently addressed all my concerns. I do not have any further questions.
Reviewer 2 Report
Comments and Suggestions for Authors
It can be accepted.
Comments on the Quality of English LanguageEnhanced.